# Recurrent Switching Dynamical Systems Models for Multiple Interacting Neural Populations

**Joshua I. Glaser**[1,2,3]**, Matthew Whiteway**[2,3]**,**
**John P. Cunningham**[1,2,3]**, Liam Paninski**[1,2,3]**, Scott W. Linderman**[4]

[1] Department of Statistics, Columbia University
[2] Center for Theoretical Neuroscience, Columbia University
[3] Zuckerman Mind Brain Behavior Institute, Columbia University
[4] Department of Statistics and Wu Tsai Neurosciences Institute, Stanford University

{j.glaser, m.whiteway, jpc2181}@columbia.edu
liam@stat.columbia.edu
scott.linderman@stanford.edu

## Abstract

Modern recording techniques can generate large-scale measurements of multiple neural populations over extended time periods. However, it remains a challenge to model non-stationary interactions between high-dimensional populations of neurons. To tackle this challenge, we develop recurrent switching linear dynamical systems models for multiple populations. Here, each high-dimensional neural population is represented by a unique set of latent variables, which evolve dynamically in time. Populations interact with each other through this low-dimensional space. We allow the nature of these interactions to change over time by using a discrete set of dynamical states. Additionally, we parameterize these discrete state transition rules to capture which neural populations are responsible for switching between interaction states. To fit the model, we use variational expectation-maximization with a structured mean-field approximation. After validating the model on simulations, we apply it to two different neural datasets: spiking activity from motor areas in a non-human primate, and calcium imaging from neurons in the nematode *C. elegans*. In both datasets, the model reveals behaviorally-relevant discrete states with unique inter-population interactions and different populations that predict transitioning between these states.

## 1 Introduction

With high-density silicon probes [1] and large-scale calcium imaging methods [2–4], neuroscientists now commonly record thousands of neurons at a time with single-cell resolution. Unlike past recording methods, which were often limited by small fields of view, sparse coverage, or low spatial resolution, these new techniques can simultaneously capture large numbers of neurons from many different brain areas and multiple cell types. With these expanding capabilities, interactions between populations of neurons can be assessed with greater precision than previously possible.

Initial multi-population studies point to three recurring themes [5]. First, though populations may consist of hundreds of neurons, the neural activity within each population may be correlated [6, 7], reflecting a lower-dimensional, latent population "state" [8], especially during simple behaviors [9] (except see [10]). Second, the interactions between populations—often quantified by the cross-correlation between neurons in one population and another—may also occur in a low-dimensional

subspace [11–13]. Third, and most importantly, these interactions are context dependent and may change over time as the subject's task-engagement, behavior, and internal state vary [14–18].

Existing models of multi-neuronal data capture some but not all of these factors. A long line of research has established state-space models for single-population analysis, exploiting the low-dimensional nature of neural firing rates observed in many (but not all) brain areas in common experimental paradigms [19–23]. Building on these state-space models, Buesing et al. [24] and Semedo et al. [25] developed probabilistic models for multiple populations of neurons, based on the assumption that the latent state of one population influences the future state of another. These multi-population models are closely connected to probabilistic canonical correlation analysis (CCA) [26], distance covariance analysis [27], and reduced rank regression [28], which find low dimensional relationships between populations.

However, these approaches assume that interactions between populations do not change over time, which may be valid in some tasks but not in general [29]. In some cases, we may know in advance when interactions are likely to change—e.g. behavioral covariates may inform switching times—but this is a strong presupposition. Ideally, we seek methods that detect changing interactions from neural data directly. Switching state space models can infer these non-stationary dynamics by introducing an additional set of discrete states that govern the dynamics at each time point [30, 31], and these methods have proven successful at modeling multi-neuronal activity within single populations [32] and multi-region dynamics in fMRI data [33]. The recurrent switching linear dynamical system (rSLDS) extends this model class to allow the low-dimensional continuous latent states to influence the switching probabilities between discrete states [34–36].

Here, we further extend the rSLDS class of models to analyze multiple interacting populations of neurons by enforcing population-specific continuous latent states and discrete interaction states. We introduce a novel parameterization of the discrete transition rules that allows the latent states of the neural populations to govern the probability of transitioning into or out of an interaction state, which enhances the interpretability of the resulting model. We fit these models with a variational expectation-maximization algorithm that leverages the Markovian dependency structure in the discrete and continuous states. We demonstrate the model's efficacy on synthetic data and two multi-neuronal datasets. In each, the multi-population rSLDS automatically detects behaviorally-relevant discrete states of neural interactions, along with the populations that predict these state changes. Code to fit these models is available at `https://github.com/lindermanlab/ssm/`.

## 2   Review of recurrent switching linear dynamical systems

The linear dynamical system (LDS) model and its variants are popular tools for modeling multi-dimensional time series. The LDS models a multi-dimensional time series using a lower dimensional latent representation of the system, which evolves over time according to linear dynamics. Switching linear dynamical system (SLDS) models extend this framework by including a set of discrete states, which are each associated with their own linear dynamics [37, 38]. The model can then switch between the different linear dynamics over time, which allows modeling nonlinear dynamical systems in a locally linear manner. The model switches are typically defined by a Markov transition matrix that specifies the probability that the system will switch from one discrete state to another (or stay the same), as in a hidden Markov model. RSLDS models further extend SLDS models by also making discrete state transitions dependent on the underlying low-dimensional latent representation, which often leads to a more accurate generative model [34, 35].

We will now describe the rSLDS model in full detail. Let $y_t$ be a vector representing the activity of $N$ neurons at time $t$; it may contain real-valued calcium fluorescence measurements or spike counts, as appropriate. Let $x_t \in \mathbb{R}^D$ denote the continuous latent state at time $t$ (often $D \ll N$). The observed activity is modeled with a generalized linear model (GLM), $\mathbb{E}[y_t] = f(Cx_t + d)$, where $C \in \mathbb{R}^{N \times D}$ and $d \in \mathbb{R}^N$ parameterize the linear mapping, and $f$ is applied element-wise to map $\mathbb{R}^N$ to the appropriate mean-parameter space. For real-valued calcium-fluorescence recordings we may use the identity mean function $f(x) = x$ and a Gaussian conditional distribution; for spike counts we use a soft-plus function $f(x) = \log(1 + e^x)$ and a Poisson distribution.

The rSLDS models the temporal dynamics of the continuous states as conditionally linear with additive Gaussian noise, given a corresponding discrete latent state $z_t \in \{1, \dots, K\}$. Each discrete

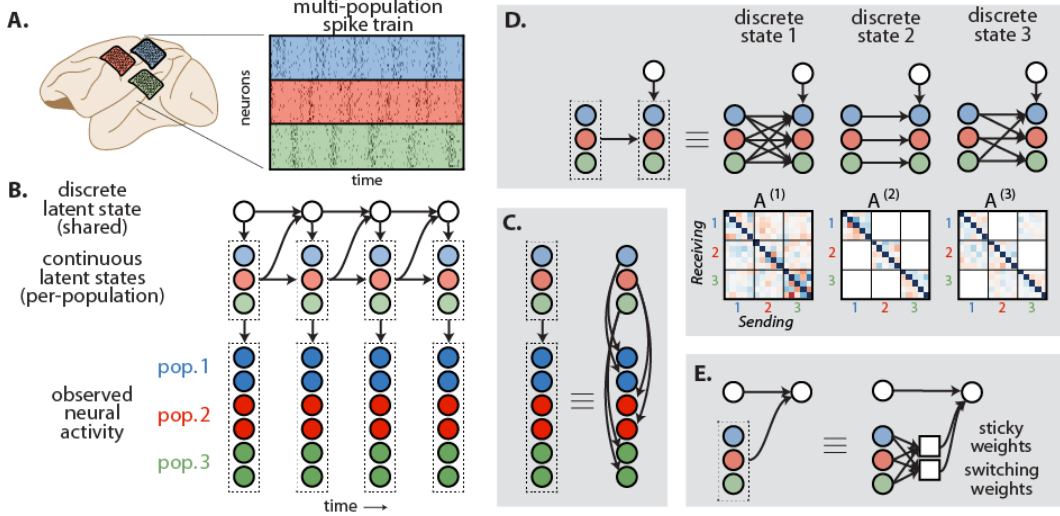

Figure 1: **A.** Illustration of a simulated multi-neuronal recording (brain image from [39]). **B.** Graphical model of the mp-srSLDS, an extension of recurrent switching linear dynamical systems to multiple interacting populations of neurons. The observed neural activity ($y$ in Eq. (3)) consists of recordings from multiple populations, denoted here by blue, red, and green. The observed activity is a function of the continuous latent state ($x$ in Eqs. (3),(4)), which also factors across populations. Continuous state dynamics are governed by a corresponding discrete latent state ($z$ in Eq. (4)). **C.** The observed neural activity in a population is a high dimensional projection of the population's continuous latent state. **D.** Each discrete state corresponds to different dynamics function, which capture different patterns of interaction between populations. **E.** Finally, the discrete state changes depend on the current population states. The continuous state determines the probability of switching discrete states (via switching weights, $R$ in Eq. (5)) or remaining in the same state (via sticky weights, $S$ in Eq. (5)).

latent state is associated with a different set of linear dynamics,

$$x_t \sim \mathcal{N}\left(A^{(z_t)} x_{t-1} + b^{(z_t)}, Q^{(z_t)}\right). \tag{1}$$

As the discrete state of the system changes, so do the corresponding linear dynamics $A^{(z_t)}$. Each discrete state is also characterized by a bias $b^{(z_t)}$ and a noise covariance $Q^{(z_t)}$.

The discrete states are modeled as a Markov process with conditional dependencies on the continuous states. In this sense, the discrete and continuous states are recurrently coupled. The discrete state at time $t$ is chosen from a categorical distribution,

$$z_t \sim \mathrm{Cat}\left(\pi_t\right), \quad \pi_t = \mathrm{softmax}(R_{z_{t-1}} x_{t-1} + r_{z_{t-1}}), \tag{2}$$

where $R_{z_{t-1}} \in \mathbb{R}^{K \times D}$ and $r_{z_{t-1}} \in \mathbb{R}^K$ parameterize a GLM that determines how the continuous latent states influence the discrete state transitions. In general, each preceding discrete state $z_{t-1}$ parameterizes its own GLM, but these parameters can be shared across discrete states (i.e. $R_k = R$ and $r_k = r$ for all $k$), as in the "shared" rSLDS [34] that we build upon.

## 3 Multi-population recurrent SLDS models

We now extend the rSLDS by considering multi-neuron activity simultaneously recorded from each of $J$ populations (Fig. 1A). Let $y_t^{(j)}$ denote a vector of activity measurements of the $N_j$ neurons in population $j$ in time bin $t$, which again may be real- or count-valued, as appropriate. We model these data as an rSLDS (Fig. 1B) with new constraints for multi-population recordings.

**Disentangling intra- and inter-population dynamics** First, we constrain the model so that each population has its own continuous latent states. Specifically, let $x_t^{(j)} \in \mathbb{R}^{D_j}$ denote a continuous

latent state of population $j$ at time $t$. The population states may differ in dimensionality $D_j$, since populations may differ in size and complexity. The observed activity of population $j$ is modeled with a generalized linear model,

$$\mathbb{E}[y_t^{(j)}] = f(C_j x_t^{(j)} + d_j), \tag{3}$$

where each population has its own linear mapping (Fig. 1C) parameterized by $\{C_j, d_j\}$. We refer to models with continuous latents separated by population using the prefix "mp", e.g. an mp-rSLDS.

Having unique continuous latents for each population allows us to decompose the dynamics in a more interpretable manner (Fig. 1D). We model the temporal dynamics of the continuous states as

$$x_t^{(j)} = A_{j \leftarrow j}^{(z_t)} x_{t-1}^{(j)} + \sum_{i \neq j} A_{j \leftarrow i}^{(z_t)} x_{t-1}^{(i)} + b_j^{(z_t)} + \epsilon_t^{(j)}, \tag{4}$$

where $\epsilon_t = (\epsilon_t^{(1)}, \ldots, \epsilon_t^{(J)}) \sim \mathcal{N}(0, Q^{(z_t)})$. The matrices $A_{j \leftarrow j}^{(z_t)} \in \mathbb{R}^{D_j \times D_j}$ and $A_{j \leftarrow i}^{(z_t)} \in \mathbb{R}^{D_j \times D_i}$ capture the within- and between-population linear dynamics, respectively. Note that $A_{j \leftarrow j}^{(k)}$ and $A_{j \leftarrow i}^{(k)}$ form the blocks of the full dynamics matrix, $A^{(k)}$ (Fig. 1D), which we display in our experiments. Note that this equation can easily be extended to include additional previous time points, e.g. $x_{t-2}$ (see Supplement). By partitioning the continuous latent states and dynamics, we can characterize the collective dynamics of the entire system in terms of contributions of its constituent populations.

**Incorporating prior information such as anatomy**  This formulation also easily admits anatomical covariates in the form of prior distributions on the model parameters. For example, it may be natural to incorporate sparsity-inducing priors on the dynamics matrices $A_{j \leftarrow i}^{(k)}$ with sparsity penalties that reflect known anatomical connectivity between brain areas, as suggested by Fig 1D.

The same idea applies when the "populations" correspond to different cell types. As a simple example, consider a two-population model with excitatory and inhibitory cells. We can model these opposing influences by constraining $A_{j \leftarrow i}^{(k)}$ to be positive or negative depending on the type of population $i$. In order for these signs to be meaningful, we also need to constrain the emission matrices $C_j$ to be non-negative. We explore these ideas further in the supplement.

**State-dependent stickiness**  Finally, we aim to create a model that allows determining whether switching or staying in a discrete state is related to neural populations' activities. To do so, we propose an alternative form of the rSLDS discrete state transitions that distinguishes between self-transitions and transitions to other states (Fig. 1E), as in the sticky hidden Markov model (HMM) [40]. To implement this stickiness, we replace the recurrent transition probabilities in eq. (2) with,

$$\pi_t = \text{softmax}\left\{ \Big( \sum_{j=1}^{J} \big(R_j x_{t-1}^{(j)}\big) + r \Big) \odot (1 - e_{z_{t-1}}) + \Big( \sum_{j=1}^{J} \big(S_j x_{t-1}^{(j)}\big) + s \Big) \odot e_{z_{t-1}} \right\}, \tag{5}$$

where $R_j, S_j \in \mathbb{R}^{K \times D_j}$, $r, s \in \mathbb{R}^K$, and $e_{z_{t-1}} \in \{0, 1\}^K$ is a one-hot encoding of $z_{t-1}$.

This formulation lies in between the shared rSLDS, which has $(K + 1)D$ parameters (here $D = \sum_j D_j$), and the full rSLDS, which has $K(K + 1)D$ parameters. The transition probabilities in eq. (5) have $2(K + 1)D$ parameters, allowing it to capture the relative probability of different discrete interaction states in different parts of continuous state space, as well as the variable stickiness of different discrete states. The "switching" and "sticky" weight matrices $R_j$ and $S_j$ (Fig 1E) capture how each population contributes to the transition probabilities—at any given time point the contributions will be $R_j x_{t-1}^{(j)}$ and $S_j x_{t-1}^{(j)}$. We call this model that incorporates state-dependent stickiness an "srSLDS", and thus an "mp-srSLDS" when used with per-population latents.

It is also possible to include a dependence on the previous discrete state, in which case (5) becomes

$$\pi_t = \text{softmax}\left\{ P_{z_{t-1}} + \Big( \sum_{j=1}^{J} R_j x_{t-1}^{(j)} \Big) \odot (1 - e_{z_{t-1}}) + \Big( \sum_{j=1}^{J} S_j x_{t-1}^{(j)} \Big) \odot e_{z_{t-1}} \right\}, \tag{6}$$

where $P_k \in \mathbb{R}^K$ represents a row of a standard transition matrix between discrete states.

## 4 Model fitting

We use the variational Laplace-EM algorithm proposed by Zoltowski et al. [41] to approximate the posterior distribution over discrete and continuous latent states and to estimate the model parameters. For completeness, we briefly describe the algorithm here. Let $\Theta$ denote the set of all model parameters. We approximate the posterior distribution on latent states given model parameters and data with a structured mean-field approximation [42],

$$
\begin{aligned}
p(z_{1:T}, \{x_{1:T}^{(j)}\}_{j=1}^J \mid \{y_{1:T}^{(j)}\}_{j=1}^J, \Theta) &\approx q(z_{1:T})\, q(\{x_{1:T}^{(j)}\}_{j=1}^J) \\
&= \left[ q(z_1) \prod_{t=2}^T q(z_t \mid z_{t-1}) \right] \left[ q(\{x_1^{(j)}\}_{j=1}^J) \prod_{t=2}^T q(\{x_t^{(j)}\}_{j=1}^J \mid \{x_{t-1}^{(j)}\}_{j=1}^J) \right]. \quad (7)
\end{aligned}
$$

This approximation leverages the conditional dependencies of the rSLDS. Specifically, as shown in previous work [41], the variational parameters can be optimized via block-coordinate ascent. The updates to the discrete state factor $q(z_{1:T})$ need only the expected log transition probabilities under the continuous state posterior approximation, and these expectations can be approximated via Monte Carlo. The update for the continuous state factor is slightly more complex; the optimal update satisfies,

$$
\begin{aligned}
\log q(\{x_{1:T}^{(j)}\}_{j=1}^J) &= \sum_{t=2}^T \mathbb{E}_{q(z_{1:T})} \left[ \log p(z_t \mid \{x_{t-1}^{(j)}\}_{j=1}^J, z_{t-1}) \right] \\
&\quad + \sum_{t=1}^T \sum_{j=1}^J \left( \mathbb{E}_{q(z_{1:T})} \left[ \log p(x_t^{(j)} \mid \{x_{t-1}^{(i)}\}_{i=1}^J, z_t) \right] + \log p(y_t^{(j)} \mid x_t^{(j)}) \right) + c. \quad (8)
\end{aligned}
$$

The continuous state dynamics terms are linear and Gaussian, but the recurrent transitions (first term) and possibly the emissions (last term) are not. However, when these probabilities are given by the generalized linear models described above, the right hand side of eq. (8) is concave and its Hessian is block-tridiagonal due to the Markovianity of the model. Thus, the mode can be efficiently obtained via first- or second-order optimization methods. We leverage these properties to form a Laplace approximation of the continuous state posterior, as in Laplace VI [43].

Finally, again following Zoltowski et al. [41], we use the posterior approximation to perform variational expectation-maximization. That is, we iterate between approximating the posterior and maximizing the expected log probability,

$$
\mathcal{L}(\Theta) = \mathbb{E}_{q(z_{1:T}) q(\{x_{1:T}^{(j)}\}_{j=1}^J)} \left[ \log p\left( z_{1:T}, \{x_{1:T}^{(j)}, y_{1:T}^{(j)}\}_{j=1}^J \right) \right]. \quad (9)
$$

The continuous dynamics parameters admit closed-form solutions in terms of expected sufficient statistics of the model, and the recurrent transition parameters and emission parameters are optimized via stochastic gradient ascent. In general, variational EM algorithms converge to local optima of the evidence lower bound (9), but it is not obvious how the variational approximation biases the learned parameters [44]. However, for recurrent SLDS, this variational Laplace-EM algorithm has been shown to be both faster and more accurate than competing methods like black box variational inference and particle EM [41].

## 5 Experiments

### 5.1 Simulations

We first tested the model on simulations of spiking neural data. We used $J = 3$ populations of neurons with $D_j = 5$ dimensional latents and $N_j = 75$ neurons per population, and $K = 3$ discrete states of dynamics, which had different interaction patterns (Fig. 2A). We let the transition probabilities of staying in a given state and switching to a given state each be based on a single neural population (Fig. 2A). Average spike rates were 0.25 per bin, corresponding to 10Hz if bins are 25 ms. In this and the following experiments, we use a simple prior that the elements of the dynamics matrices are independent Gaussians with $\mathbb{E}[A^{(k)}] = I$, whose strength is determined on held out data.

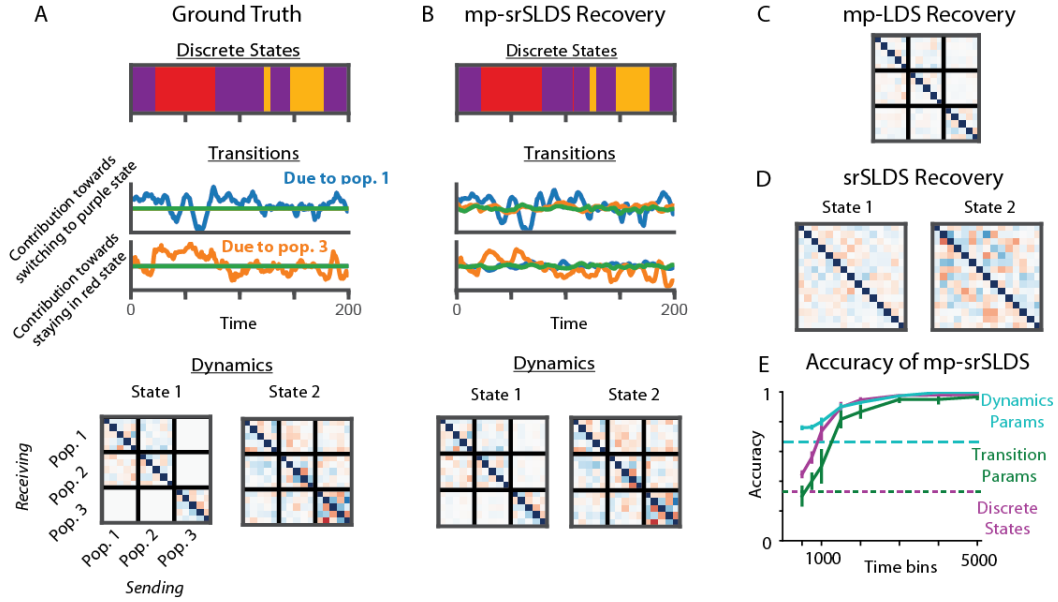

Figure 2: *Simulations.* **A.** Ground truth of a multipopulation srSLDS simulation with 3 discrete states (top). The effect of each population $j$ on the transition probabilities is $R_j x^{(j)}$ and $S_j x^{(j)}$, for switching and staying in states, respectively (middle). In this simulation, we let the probabilities of staying in a given state and switching to another given state each be based on a single neural population (middle). Different discrete states have different connectivity patterns in their dynamics (shown for two states; bottom). **B.** The recovery of discrete states, dynamics, and transitions parameters using an mp-srSLDS model. For a clearer comparison when plotting, each population's estimated continuous latents (which can be arbitrarily linearly modified) were separately aligned to the simulation ground truth, and the dynamics matrix was updated accordingly. Note that this does not alter the underlying block structure of the recovered dynamics. **C.** The recovered dynamics in an mp-LDS model (latents divided by population, but only one discrete state). **D.** The recovered dynamics in an srSLDS model, where the latents are not separated by population. **E.** The accuracy of an mp-srSLDS model as a function of the number of time bins. For the accuracy of transition parameters, we calculate the population that has the most influence on staying in each state and switching to each state, and determine whether these match ground truth. To determine the accuracy of dynamics parameters, we assess the sparsity of the inferred connections (block zeros in the dynamics matrix, as in A). To calculate a metric, we consider a block to be zero if its mean values are less than 25% of the mean of the largest block. Dashed lines represent chance level for each metric. Error bars are standard errors across simulations.

In these simulations, the mp-srSLDS models with separate latents per population could accurately capture changing dynamics/interactions over time, as well as the populations responsible for discrete state transitions (Fig. 2B). As would be expected, an mp-LDS model with stationary dynamics failed to estimate the interactions; here it concluded that there were weaker interactions at all time points, as opposed to some time points with no interaction and some time points with a stronger interaction (Fig. 2C). Additionally, an srSLDS model, which did not separate the latents by population, recovered dynamics that could not easily be interpreted in terms of population-to-population interactions (Fig. 2D). This demonstrates the importance of the per-population latent constraints.

For quantitative metrics, we tested the models' ability to determine the discrete states, the interactions within each discrete state, and the populations responsible for the discrete state transitions (see Fig. 2 caption). In these simulations, the model accurately recovered ground truth parameters within $\sim$3000 time bins (on the order of minutes of data), an experimentally reasonable amount (Fig. 2E).

## 5.2 Array recordings in motor and premotor cortex

We next analyzed a dataset in which spiking activity was simultaneously recorded from dorsal premotor (PMd) and primary motor (M1) cortices while a monkey made reaching movements [45]. In the experiment, two potential reach targets were shown, followed by a delay period. Later, one

of the targets was shown again, at which point the monkey reached to that target. We model the time period from delay start to reach end. Spikes from 73 M1 and 107 PMd neurons were counted in 25ms bins, resulting in over 13000 time bins across over 200 trials. We used $K = 4$ discrete states and $D_j = 10$ dimensional latents for each population, as determined by cross-validation. For our model of discrete state transitions, we used Eq. (6) rather than (5), due to the consistent task structure.

First, we show that the added features of our model that aid in interpretability—namely separate latents per population and a sticky recurrent parameterization—did not sacrifice performance. To evaluate how well the models could predict future neural data, we inferred the latents $x_t$ and $z_t$ using preceding data, and then predicted into the future using the learned generative model. The sticky recurrent parameterization slightly improves performance (Fig. 3G). Constraining the model to have independent latents per population gives almost the same future prediction likelihoods as an unconstrained model (Fig. 3H). Additionally, we show better future performance compared to independent models of M1 and PMd (Fig. 3H, green), showing that the model's inter-region interactions are useful for future predictions.

Next, we investigated the parameters of the fitted mp-srSLDS model. The inferred discrete states roughly correspond to the delay and movement epochs (Fig. 3B), along with a third, intermediate state. In the model, PMd primarily predicts switching out of the delay epoch and into the movement epoch (Fig. 3C). The dynamics matrices from those two discrete states show clear differences (Fig. 3D). Using these dynamics matrices, we can investigate information transmission between populations. For example, we see that there is a greater influence of external states during the movement period. Specifically, the influence of M1 on PMd, relative to its internal dynamics, is greater during movement (Fig. 3E), likely reflecting the known feedback connections. We also see that the projection from one region to another has lower dimensionality than their internal dynamics (Fig. 3F), supporting the hypothesis of lower dimensional shared subspaces between neural populations [11].

## 5.3   C. elegans calcium imaging

Finally, we demonstrate the multi-population srSLDS in a dataset with different notions of population structure—a calcium recording in the nematode *C. elegans* [47]. The worm has 302 neurons, each with a unique label (e.g. AVAL), and most neurons come in lateral pairs (denoted by an 'L' or 'R' suffix in the label). In the experiment, worms were immobilized and calcium fluorescence was measured at 3Hz in approximately 100 neurons per worm. We analyzed 18 minutes ($\sim 3250$ time bins) of activity from one worm, in which 48 neurons could be confidently labeled. A subset of the data is shown in Fig. 4A. In addition to the neural activity, the dataset also contains annotations of fictive behaviors based on the neural activity.

There are many ways of defining neural populations in *C. elegans*. Here, we group correlated lateral pairs into populations of size two, and group the unlabeled neurons into their own "population." The final model has $D = 37$ dimensional continuous latent variables (26 individual neurons and 11 pairs). There are many reasonable definitions of population in *C. elegans*; e.g. we consider neurons grouped by ganglia in the supplement.

As in the preceding analysis, we first made quantitative comparisons. We confirmed that the multipopulation model is able to predict future activity as well as a model with unconstrained latents (Fig. 4G). Additionally, our (more interpretable) model with sticky transitions predicts future activity as well as a model with standard recurrent transitions, and outperforms models without recurrent transitions (Fig. 4F). We used $K = 10$ discrete states and then analyzed the parameters of the fitted model for evidence of sparse and behaviorally-relevant interactions.

The model uncovered discrete states that share a reasonable amount of overlap with the manually annotated behavioral states (Fig. 4D). Interestingly, the model finds a small number of neurons that predict the transitions. For example, the neuron pair SMDD(R/L), which is known to be involved in dorsal turns [47], predicts switching into a state overlapping with the annotated dorsal turn (Fig. 4C). There are also different patterns of functional interactions in the different states (Fig. 4E). For instance, during a state corresponding to ventral turns, the pair SMDV(R/L), which is known to be involved in ventral turns [47], receive greater input. Thus, the model can help to uncover small numbers of neurons involved in behaviorally-relevant states.

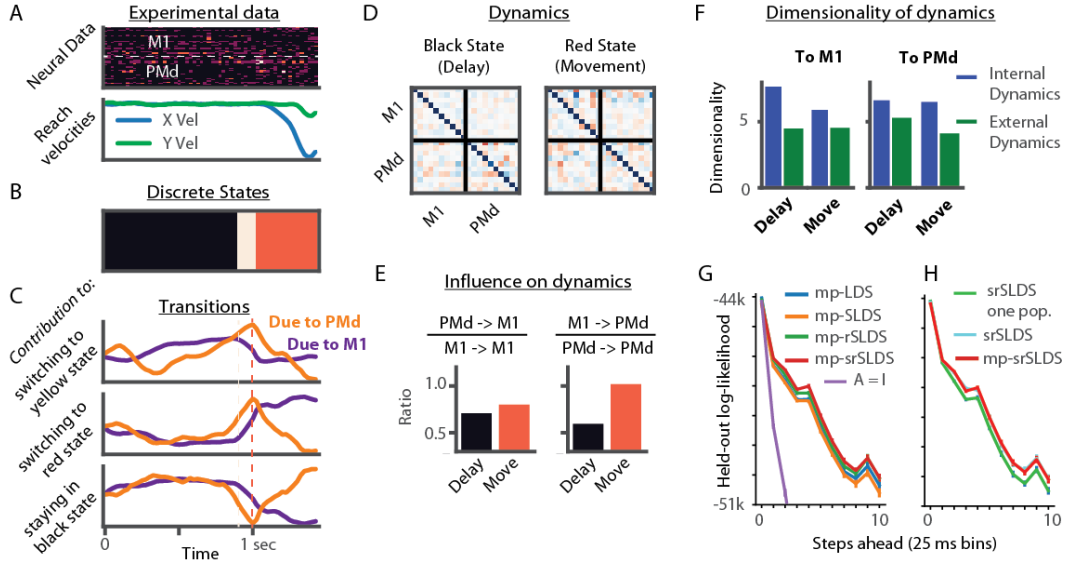

Figure 3: *Multi-population model of data from non-human primate premotor and motor cortices.* **A.** Data from an example trial. Neural data (top) is shown from an example 50 neurons – the top half from M1 and bottom half from PMd. Reach velocities are below. **B-E** Results from fitting an mp-srSLDS model with 4 discrete states. **B.** Discrete states from an example trial. **C.** The contributions of M1 and PMd towards staying in, or switching to, some of the discrete states, calculated as in Fig. 2. **D.** Learned dynamics matrices for two of the discrete states, that roughly correspond to delay and movement periods. **E-F.** Using the dynamics, $A^{(z_t)}$, from panel D, we can determine the change in the continuous latent state due to internal dynamics (e.g. M1 to M1) as $(A^{(z_t)}_{j \leftarrow j} - \mathbb{I})x^{(j)}$ or external dynamics as $(A^{(z_t)}_{j \leftarrow i})x^{(i)}$. **E.** The ratio of the magnitude of external to internal contributions. **F.** The dimensionality of the internal and external dynamics, calculated as $\left(\sum \lambda\right)^2 / \left(\sum \lambda^2\right)$, where $\lambda$ are the singular values, as in [46]. **G.** The accuracy of 5 models at making future predictions up to 10 time bins ahead on held-out data (1/4 of trials). Log likelihood of the activity is shown based on the estimated Poisson rates. The purple trace signifies a baseline with identity dynamics, i.e. a random walk process. **H.** The mp-srSLDS is constrained to have separate latent variables for each population (red). We compare this to the unconstrained model with latents shared between populations (cyan; overlaps with red), and to srSLDS models fit to independent populations (green).

## 6   Discussion

We introduced a recurrent switching dynamical systems model to facilitate the analysis of multiple interacting neural populations (mp-srSLDS). This work addresses the challenging and open problem of modeling interactions between high-dimensional neural populations that may be changing over time in an unknown manner. It builds upon previous work on recurrent switching models for single population neural data [34–36, 41] and non-switching models for multi-population neural data [24, 25]. Furthermore, the mp-srSLDS incorporates several constraints that allow for easier interpretation of the resulting model, including distinctions between intra- and inter-population dynamics, the incorporation of anatomical side information if available, and a parameterization of discrete state transitions that distinguishes activity leading to self-transitions from activity leading to transitions out of a state. We demonstrated the increased efficacy and interpretability of this model on simulated and real neural data.

Though the mp-srSLDS model is designed to illuminate how multiple neural populations interact with one another, we must also be clear that the model should not be interpreted as inferring the causal effects occurring within the brain. In most experimental paradigms it is impossible to record activity from every neuron, and thus there will be many hidden variables (other populations) that may affect activity in the recorded populations. Including continuous latent states to account for unmeasured data is a direction for future work. Another reason for inaccurate predictions could be model mismatch; for example, the relation between the continuous latents and neural activity could

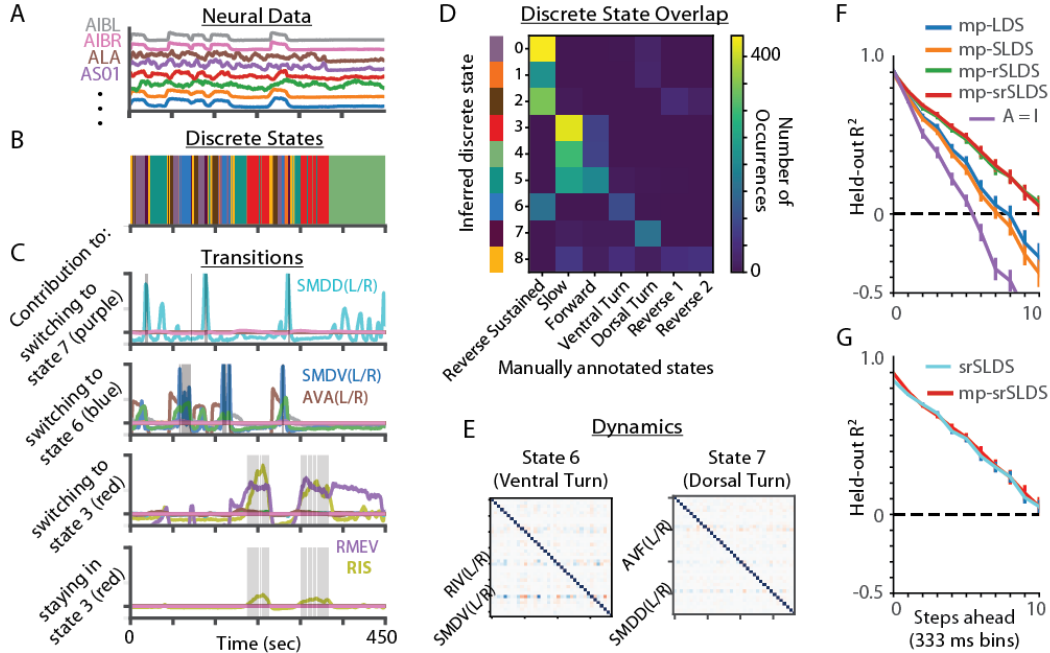

Figure 4: *Multi-population model of C. elegans calcium imaging data.* **A.** Calcium imaging data from example neurons (in different colors). **B-E.** States and parameters of a fitted mp-srSLDS. **B.** The recovered discrete states for an example time period. **C.** The contributions of various neurons or neuron groups towards staying in, or switching to, some of the discrete states. The relevant discrete state for the given panel is overlaid in gray. **D.** The overlap of the inferred states with manually annotated states. Colors on the left match with the discrete states in panel B. **E.** Learned dynamics matrices for 2 of the discrete states. **F.** The accuracy of 5 models at making future predictions up to 10 time bins ahead. Here we show the mean $R^2$ across neurons on a held-out 10% of the data. **G.** The comparison of the mp-srSLDS model to an unconstrained srSLDS model. Error bars are standard errors across generative samples in panels F and G, along with in Fig. 3G and H.

be nonlinear, neural activity may not follow Poisson statistics, or there may be different discrete states underlying internal and external dynamics. By incorporating previous research on these topics, such as on nonlinear emissions [23], general count LDS models [48], and factorial models [49], future work can strive towards a more complete model of multiple interacting neural populations.

## Broader Impact

Understanding neural computation, and the interaction between multiple brain regions, is critical for better treating neurological disorders. Here, we develop an analysis tool that can help to understand multiple populations of neurons, and aims to make progress towards this long-term goal. However, this is a decades-long effort and we foresee no immediate societal consequences of this work.

## Acknowledgments and Disclosure of Funding

We thank Brian Dekleva and Lee Miller for sharing non-human primate motor cortical data. We thank Manuel Zimmer for sharing *C. elegans* data.

J.I.G., M.R.W., J.P.C. and L.P. were supported by grants from the Simons Foundation (542963 and 543023), the Gatsby Charitable Foundation (GAT3708), and the NSF NeuroNex Award (DBI-1707398). J.I.G. was also supported by NIH (K99NS119787). S.W.L. was supported by grants from the Simons Collaboration on the Global Brain (SCGB 697092) and the NIH BRAIN Initiative (U19NS113201 and R01NS113119).

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
