[Supplementary Material]

# Supplement: Recurrent Switching Dynamical Systems Models for Multiple Interacting Neural Populations

**Joshua I. Glaser**[1,2,3], **Matthew Whiteway**[2,3],
**John P. Cunningham**[1,2,3], **Liam Paninski**[1,2,3], **Scott W. Linderman**[4]

[1] Department of Statistics, Columbia University
[2] Center for Theoretical Neuroscience, Columbia University
[3] Zuckerman Mind Brain Behavior Institute, Columbia University
[4] Department of Statistics and Wu Tsai Neurosciences Institute, Stanford University

{j.glaser, m.whiteway, jpc2181}@columbia.edu
liam@stat.columbia.edu
scott.linderman@stanford.edu

## A  Bayesian spike-and-slab regression

As discussed in the main text, there could be many ways to incorporate anatomical priors into our formulation. Here, we demonstrate one example—assuming that brain regions are sparsely connected, and therefore many blocks of the dynamics matrices will be zero. This can be implemented using a block-wise spike-and-slab prior on the dynamics matrices.

### A.1  Formulation

We want a block-sparse prior for dynamics matrices. For simplicity, let us assume we have the dynamics matrix $A \in \mathbb{R}^{D \times D}$ for a single discrete state. We break down this matrix into a $B \times B$ set of blocks, where $B$ is the number of neural populations. We place a spike-and-slab prior on the weights such that entire blocks are exactly zero with some probability. For simplicity, assume that the blocks are all the same size $S_{\mathsf{out}} \times S_{\mathsf{in}}$, where $S_{\mathsf{out}} = S_{\mathsf{in}} = D/B$ (i.e., each population has the same number of continuous latents). Let $\tilde{M} \in \{0,1\}^{B \times B}$ be a binary matrix indicating whether a corresponding block is zero or not. The full model is,

$$
\begin{aligned}
x_{t+1} &\sim \mathcal{N}((A \odot M)x_t, \sigma^2 I) & \text{for } t \in 1,\ldots,T \\
a_{ij} &\sim \mathcal{N}(0, \eta^2) & \text{for } i \in 1,\ldots,D; j \in 1,\ldots,D \\
M &= \tilde{M} \otimes 1_{S_{\mathsf{out}} \times S_{\mathsf{in}}} \\
\tilde{m}_{k,\ell} &\sim \mathrm{Bern}(\rho) & \text{for } k \in 1,\ldots,B; \ell \in 1,\ldots,B.
\end{aligned}
$$

We would like to approximate the posterior distribution $p(A, \tilde{M} \mid \mathcal{D})$, and particularly its mode. First consider the posterior of $A$ for a fixed sparsity pattern,

$$p(A \mid \tilde{M}, \mathcal{D}) \propto p(A) \prod_{t=1}^{T} p(x_{t+1} \mid A, \tilde{M}, x_t)$$

$$= \prod_{i=1}^{D} \left[ \mathcal{N}(a_i \mid 0, \eta^2 I) \prod_{t=1}^{T} \mathcal{N}(x_{t+1,i} \mid (a_i \odot m_i) x_t, \sigma^2) \right]$$

$$= \prod_{i=1}^{D} c_i \, \mathcal{N}(a_i \mid J_i^{-1} h_i, J_i^{-1}),$$

where

$$J_i = \frac{1}{\eta^2} I + \frac{1}{\sigma^2} \sum_{t=1}^{T} (x_t x_t^\top) \odot (m_i m_i^\top)$$

$$h_i = \frac{1}{\sigma^2} \sum_{t=1}^{T} x_{t+1,i} (x_t \odot m_i)$$

$$c_i = \frac{\exp\left\{ -\frac{1}{2\sigma^2} \sum_{t=1}^{T} x_{t+1,i}^2 \right\}}{(2\pi\sigma^2)^{T/2} \eta^D} |J_i|^{-1/2} \exp\left\{ \frac{1}{2} h_i^\top J_i^{-1} h_i \right\}$$

The posterior marginal of $\tilde{M}$ is given by,

$$p(\tilde{M} \mid \mathcal{D})$$

$$\propto p(\tilde{M}) \int p(A) \prod_{t=1}^{T} p(x_{t+1} \mid A, \tilde{M}, x_t) \, \mathrm{d}A$$

$$= p(\tilde{M}) \prod_{i=1}^{D} \int \mathcal{N}(a_i \mid 0, \eta^2 I) \prod_{t=1}^{T} \mathcal{N}(x_{t+1,i} \mid (a_i \odot m_i) x_t, \sigma^2) \, \mathrm{d}a_i$$

$$\propto p(\tilde{M}) \prod_{i=1}^{D} |J_i|^{-1/2} \exp\left\{ \frac{1}{2} h_i^\top J_i^{-1} h_i \right\}$$

$$= \prod_{k=1}^{B} \left[ \prod_{\ell=1}^{B} \mathrm{Bern}(\tilde{m}_{k,\ell} \mid \rho) \prod_{i=(k-1)S_{\mathrm{out}}+1}^{kS_{\mathrm{out}}} |J_i|^{-1/2} \exp\left\{ \frac{1}{2} h_i^\top J_i^{-1} h_i \right\} \right].$$

The posterior factorizes over rows of $\tilde{M}$ so that we don't have to sum over $2^{B^2}$ binary assignments, only $2^B$ assignments for each row of $\tilde{M}$. Still, if there are too many blocks to enumerate, we can take a mean field approach instead.

Once we have the exact or approximate posterior, we can find the most likely sparsity pattern $\tilde{M}^\star = \arg\max p(\tilde{M} \mid \mathcal{D})$. Given $\tilde{M}^\star$, the optimal regression weights are at the mean $J_i^{-1} h_i$ of the conditional distribution of $A$, given above.

## A.2   Experiments

To show the potential of this approach, we ran a simple set of simulations. Like the simulations in the main text, we let there be 3 populations with 5 continuous latents each, and 75 neurons per population. The average firing rate per neuron was 0.25 spikes per bin. Here, we let there be a single discrete state.

We evaluated the ability of spike-and-slab models to recover the sparse block structure, versus the standard prior used in the main text. As a reminder, the standard prior is that the elements of the dynamics matrices are independent Gaussians with $\mathbb{E}[A^{(k)}] = I$. This can also be thought of as an L2 penalty on the matrix $A^{(k)} - I$. The spike-and-slab model directly outputs an estimate of the

Figure S1: *Simulations with a block-wise spike-and-slab prior.* **A.** The ground truth dynamics matrix (left), along with the estimated dynamics parameters given the block-wise spike-and-slab prior (middle) and the standard prior (right) that the elements of the dynamics matrices are independent Gaussians with $\mathbb{E}[A^{(k)}] = I$. **B.** The accuracy of inferring the sparsity structure of the dynamics matrices, for differing numbers of time bins. For the standard prior, as in Fig. 2, we consider a block to be zero if the block's mean value is less than 25% of the the largest block's mean. The dashed line represents chance performance, given that 1/3 of blocks were set as zero on average. Error bars are standard errors across simulations.

most likely binary sparsity mask, $\tilde{M}^\star$. However, the standard model does not. Thus, as in Fig. 2, for the standard prior, we consider a block to be zero if the block's mean value is less than 25% of the mean of the largest block. For the spike-and-slab model, we set the hyperparameter $\eta$ to be the standard deviation of the elements of $A^{(k)} - I$ calculated using a standard regression. We let the expected sparsity, $\rho$, be 0.5. When comparing these priors, we find that for limited amounts of data, this spike-and-slab prior increases the ability to accurately determine group-wise sparsity (Fig. S1).

## B   Dynamics with multiple lags

In the main text, we considered dynamics of the form:

$$x_t^{(j)} = A_{j \leftarrow j}^{(z_t)} x_{t-1}^{(j)} + \sum_{i \neq j} A_{j \leftarrow i}^{(z_t)} x_{t-1}^{(i)} + b_j^{(z_t)} + \epsilon_t^{(j)}, \tag{1}$$

where $\epsilon_t = (\epsilon_t^{(1)}, \ldots, \epsilon_t^{(J)}) \sim \mathcal{N}(0, Q^{(z_t)})$. The matrices $A_{j \leftarrow j}^{(z_t)} \in \mathbb{R}^{D_j \times D_j}$ and $A_{j \leftarrow i}^{(z_t)} \in \mathbb{R}^{D_j \times D_i}$ capture the within- and between-population linear dynamics, respectively. In this equation, the continuous latents are only dependent on the continuous latents at the previous time step (an AR(1) model). Within our framework, it is easy to extend the dynamics to be dependent on multiple previous time steps (an AR(p) model):

$$x_t^{(j)} = \sum_{\tau=1}^{p} \left[ A_{\tau, j \leftarrow j}^{(z_t)} x_{t-\tau}^{(j)} + \sum_{i \neq j} A_{\tau, j \leftarrow i}^{(z_t)} x_{t-\tau}^{(i)} \right] + b_j^{(z_t)} + \epsilon_t^{(j)}, \tag{2}$$

where $p$ is the maximum lag considered.

We created a simulation example in which internal dynamics occur with a lag of one timestep, but external dynamics occur with a lag of two timesteps (Fig. S2A). This was the case in both discrete states. The simulation had 3 population of 200 neurons each, 10000 time points, and an average firing rate of 1 spike per bin. In this simulation, we were able to correctly recover, in both discrete states, that internal dynamics primarily occurred at a time lag of one step and external dynamics at a time lag of two steps (Fig. S2B).

To achieve success in distinguishing the effects of dynamics at lag 1 versus lag 2, this example required additional data and higher firing rates than in the AR(1) case. This is likely due to the high correlation between subsequent time bins of neural activity. In the future, it could be advantageous to incorporate the spike-and-slab prior (Supplement A) to be able to accurately determine the lag time of effects, especially when many potential lags are considered.

Figure S2: *Simulation with multiple lags* **A.** Ground truth of a multipopulation srSLDS simulation with lags of up to 2 time bins, and 2 discrete states (top). In this simulation, internal dynamics (block-diagonal elements of the dynamics matrix) occur with a lag of one timestep, but external dynamics occur with a lag of two timesteps. **B.** The recovery of discrete states and dynamics parameters using an mp-srSLDS model. For a clearer comparison when plotting, each population's estimated continuous latents (which can be arbitrarily linearly modified) were separately aligned to the simulation ground truth, and the dynamics matrix was updated accordingly. Note that this does not alter the underlying block structure of the recovered dynamics.

Additionally, we note that at the current time we have fit this multi-lag model with an additional approximation: here the posterior of the continuous state has the covariance structure of an AR(1) model. In the future, we plan to cast the AR(p) model as an AR(1) model on a higher dimensional state, which will allow proper and efficient use of the variational Laplace-EM algorithm.

## C  Multi-population *C. elegans* modeling

There could be many ways of dividing the *C. elegans* into neural populations. Here, we show the results for another way of segmenting the data – by ganglia (Fig. S3A). Similar to the results in the main text, discrete states share a large overlap with the manually annotated states (Fig. S3D), individual ganglia are often responsible for transitions between discrete states (Fig. S3C), and different ganglia are primarily driven (receive the majority of the dynamics input) during different discrete states (Fig. S3E).

Figure S3: *Data from C. elegans calcium imaging.* **A.** Calcium imaging data from all labeled recorded neurons, from 5 ganglia. **B-E.** States and parameters of a fitted mp-srSLDS. **B.** The recovered discrete states for an example time period. Note that the colors do not match those in the main text. **C.** The contributions of various ganglia towards staying in, or switching to, some of the discrete states. The relevant discrete state for the given panel is overlaid in gray. Note that in this model, transitions were primarily due to the parameters involved in switching between states, rather than staying in states. **D.** The overlap of the inferred states with manually annotated states. Colors on the left match with the discrete states in panel B. **E.** Learned dynamics matrices for two of the discrete states.