[Reviews · NeurIPS 2020]

Review 1

Summary and Contributions: [Post author reply:] I'm satisfied with the authors' comments and commend them for their great work. re: latency---It's not obvious to me why considering AR(p) will allow us to identify lags in the activity of different populations. Wouldn't you rather allow for y_t to read out from more time bins (Eqn 3), y_t = f(c x_{t,t-1,t-2,...} + d)? I suggest you make this clear in the Appendix/where ever you talk about latency. ----- This paper proposes a dimensionality reduction technique to analyze neural population activity from two or more brain areas. This topic is of high interest to the computational neuroscience community, as new datasets are being released with this kind of data. This work extends a nice line of research about switching linear dynamic systems---on one hand more expressive than a single linear dynamical system but on the other hand more interpretable than an RNN. The authors argue that this framework can be posed for multiple brain regions by assuming interactions between brain regions also change based on different states. For example, in state 1, a sensory brain area may convey information to a downstream decision area while the animal integrates evidence. However, in state 2, the roles may reverse: the downstream area may send feedback to the sensory brain area to clean up/bias the sensory processing. Overall, the authors present a concise, expressive, interpretable, robust method for uncovering these dynamics.

Strengths: The authors should be proud. I thought the work was excellent, well-motivated, and will build a great foundation for future multi-area dim reduction methods. The author's work is very timely for the field and will be of great interest to the NeurIPS and neuroscience communities. The authors not only check their method with simulations, but also apply their method to two separate datasets (one in monkeys, one in worms), demonstrating the general usefulness for their method. The figures and text were also clear and straightforward. In sum, well done! I give the work a 7. I would have given it an 8, except that the method is a little incremental (very similar to the rSLDS model, but in a blocked structure) and the authors did not consider latency, which is a crucial aspect of inter-area interactions, in their model.

Weaknesses: I found no major weaknesses with the paper. The only main question I had is that it appears your method cannot detect latencies between brain areas. For example, in Fig 3E, we can see which interactions are stronger for the different states. However, is your method missing possible shared latents that are *shifted* in time? Your method would soak these up in the M1->M1, PMd->PMd dynamics. This is important because latencies (e.g., V2 lags V1 by 5ms) gives us a sense of causality, and at the end of the day, neuroscientists will want to know how information flows from one brain area to another. I do not think you need to add latency or address this with analyses in your paper, but you should have some sentences about how your method can/cannot detect latency.

Correctness: I have no concerns about the correctness of the work or the reported results. However, I would advise the authors to add more intuition for the following: - Fig 2B 'Dynamics', it's not a great match between the recovered dynamics matrices (eg. state 2 is orange on upper triangle on left, but mostly blue on right). Why is this the case? In general, give a sense of which parameters are most robust for estimation versus hardest to estimate given finite data. - Fig 3C, what are the big changes after movement onset for the red and black states (purple above orange, and orange above purple)? What does that mean? - lines 220-226, Can you add one last sentence that has a major science takeaway/prediction that you found with your analyses? E.g,. "Overall, our analyses, powered by our method, suggests that only a few neurons are responsible for transitions." It helps the reader realize how great your method can be at doing science.

Clarity: The paper was well-written and the figures were great--bravo! Only a few suggestions/comments: - lines 119-131: Better motivate why you think including a sticky prior is appropriate for neural data. This was not clear to me. Do you think it is more appropriate for neural data? Is it just to get better estimates? Reduce computation time? Say this up front in the paragraph. - Fig. 3D, what is the directionality for the plot? Is it y-axis to x-axis (so top right is M1-->PMd)? Also, not clear what boxes in D are being used to compute E (I assume it's some ratio between the mean values of the boxes). - lines 212-25, clearly state in text/caption the total # of populations you are considering. D=37 dimensions does not tell me this info. It's two neurons/population for left/right neurons and then 1 neuron/population for any remaining neurons? - Fig 1A, the coloring is confusing. To the lay reader, it seems you are only recording from one brain area, and you separated the activity into three groups. I suggest (since it's fake data) to have three separate brain areas highlighted, each one with an arrow to a different group of neurons in the population raster.

Relation to Prior Work: The paper does a good job at giving an overview of where the field is currently at. A few comments: - line 27 "neural activity within each population may be highly correlated (except see [6]"...unclear why you are referencing 6 here, which is for repeat-averaged responses, and still shows high correlation (they show neural activity follows a power law, so the first dimension captures a ton of shared stimulus variability). I would instead suggest citing Stringer et al., 2019 "Spontaneous behaviors drive multidimensional, brainwide activity" (and Musall et al., 2018 "Movement-related activity dominates cortex during sensory-guided decision making") which bolster your case that a bunch of areas are interacting together with all these global signals. I would also remove "highly" here --- noise correlations are typically 0.05 to 0.1. In your work, you are primarily considering noise correlations (trial-to-trial variability), not signal correlations (different reach directions). - in lines 38-39, likely you want to reference "Distance Covariance Analysis" from Byron Yu's lab as well, I think it does dim reduction for multiple brain areas. There may also be recent work from the Pillow lab? But not sure. How long does your algorithm take to process data, vs. Buesing + Semedo's work? Maybe just say the running times for your monkey + worm dataset. This is helpful for practitioners who have tradeoffs of time vs. accuracy. It would be nice to include some result not with LDS (such as reduced rank regression), but I suppose that's for another day.

Reproducibility: Yes

Additional Feedback: A few comments while reading: - line 68: rSLDS or RSLDS? - Check references. E.g. in [7] you have "M Yu Byron" but it's "Byron M Yu". Likely a google scholar mistake. - line 190, I found this sentence confusing. I don't think you are constraining the model---with srSLDS you give no brain area information, so in some sense it's the best possible (latents can be independent or shared). I guess I don't know what you are comparing it to for the "same" likelihoods. - put your "code available" statement either as a footnote in the intro or in the discussion. Readers who want to find that info quickly don't want to read page 5 to get it.


Review 2

Summary and Contributions: In this paper, the authors introduce a multi-population extension of a popular, switching-state-space model of the activity of neurons. They explore some variants of the model that parameterize the continuous and discrate states and interactions in more or less rich ways. They apply the model to simulated data, and data from macaque motor and premotor cortex, and c.elegans.

Strengths: The method is a very sensible extension of existing models in a direction that is inreasingly important (namely multiple populations). It exploits existing techniques for things like structured variational inference, to realize a useful algorithm. It performs extremely well on simulated data; the conclusions on the 'real' data are perhaps a little harder to interpret.

Weaknesses: I was a bit disappointed at the rather limited use of simulated to show more about when we can expect the method to work and struggle. For instance, the inference algorithm contains some notable approximations (for instance in the formal decoupling of the influence of switching); it would have been good to have seen the consequenes of this laid out more clearly. I was also a bit disappointed at the analysis of the real data. In particular, since it seems that structurally simpler models do pretty much just as well in terms of prediction, the paper declares that the benefit of their preferred model is one of interpretabbility. However, not much real insight seemed to be generated. In the macaque example, for instance, the apparent influence seems to be backwards from natural expectations (one would think that PMd would influence M1 to move; not M1 influence PMd *during* movement). The paper notes that there one cannot interpret the model in causal terms anyhow (tempting though that might be) - so what is the nature of this exess interpretability?

Correctness: Yes - I thikn the method is correct - although, as noted, in need of some compelling failure mode testing.

Clarity: Yes - the paper is very well written - albeit suffering from a surfeit of acronyms.

Relation to Prior Work: Yes - the relationship with prior work is suitably described - and the prior methds are often compared. They don't perform that much worse.

Reproducibility: Yes

Additional Feedback: [Post author reply]: I was asking for some more interpretation - this is a difficult question, but I don't think that 'clear differences' really consistutes interpretability. If the authors derived insight into the underlying algorithm associated with these areas, they didn't really share it.


Review 3

Summary and Contributions: This paper extends the family of Switching Linear Dynamical Systems to a multi-population setting, and proposes a fitting procedure to infer parameters from data. The model and inference process are clearly exposed. The model is tested on simulated data, a motor control dataset and a C elegans dataset.

Strengths: - clear modelling framework - principled fitting algorithm for inferring parameters - applications to two neuroscience datasets - highly relevant for the NeurIPS community

Weaknesses: - It is unclear how the newly introduced multi-population extension of rSLDS models is different from simply increasing the number of latent states, and imposing a block-like prior on the C matrices. This strongly limits the novelty. Moreover the constraint that different areas a priori correspond to different latent variables seems ad-hoc and unjustified. - the benefit of the new model class for modelling data is not obvious from the shown examples. This is in particular the case for the C.elegans dataset (fig 4g) where the fit leads to on the order of a hundred populations, with one or 2 neurons. - altogether rather incremental

Correctness: yes.

Clarity: Overall very clear. Fig 1: it would be helpful to add labels x,y,z next to relevant parts of the diagrams. Fig 2A-B, middle panels: not sure what is plotted

Relation to Prior Work: discussed extensively.

Reproducibility: Yes

Additional Feedback: In Fig 2, log-likelihoods for different models, similar to those computed on the neural datasets, could be shown.


Review 4

Summary and Contributions: The paper addresses the problem of quantifying the non-stationary interactions between a set of neural populations. Studying changing interactions especially for complex interacting systems is a very challenging problem and their main contribution to the existing body of literature (Semedoetal) is to add the recurrent switching dynamical system which makes it possible to capture time-dependent interactions.

Strengths: The method is well developed and has enough theoretical ground. They tested the method on simulations and real data.

Weaknesses: Some of the assumptions beyond the applicability of this method could be explained better especially the limitations and interpretability of the switching dynamics. Also, since the interaction between the populations of neurons is reduced to the interactions in the latent space, it is not clear what is the consequence of using the mean-field approximation on the interaction structure of the data. [edit: their answer on the mean-filed actually made me more confused and I feel that needs better explanation or justification]

Correctness: The methodology is well written. There is no reason to question the correctness of the method.

Clarity: The general method is clear but some clarifications about the limitations imposed and the assumptions in the method would be useful specially in using the method on real data. What is the time scale of the change in the model and how it is selected for a particular data and how much that can affect the results.

Relation to Prior Work: The paper gives enough referernces to the prior works and how it is related to them. What is missing is a better numerical comparison between what their method predicts with respect to other methods both in terms of fitting performance and in terms of switch detection. [edit: I am not convinced with their answer on the novelty issue.]

Reproducibility: Yes

Additional Feedback:

[Author Response · NeurIPS 2020]

We would like to thank all the reviewers for their thoughtful comments and their enthusiasm for our work. The reviewers' primary questions and concerns can be roughly condensed into four categories, which we address below.

**Time resolution and latency between areas (R1, R4).** These are very important considerations for maximizing the utility of our method for the neuroscience community. For simplicity, we only considered AR(1) models (i.e. $x_{t+1} = f(x_t)$), but the mp-srSLDS easily extends to AR($p$) dynamics where many previous time bins are considered. By using AR($p$) dynamics with small time bins, we can determine the latency of effects between populations. In fact, we already have an option for using AR($p$) dynamics in our code, and will include an example in the Appendix. We would also like to mention that in our analyses of neural data, the interaction trends we observed were robust to varying time bin sizes (we tested 10, 25, and 50 ms in the motor cortex data). This may be because when using an AR(1) model, the dynamics may implicitly take more time scales into account by using additional latent dimensions to integrate over time.

**Consequences of approximate posterior inference (R2, R4).** How does the structured mean-field posterior approximation $p(z, x \mid y) \approx q(z)q(x)$ affect the inferred states and learned parameters? We found that Laplace EM with this mean-field approximation outperformed standard black box variational inference in the collapsed model obtained by summing over discrete states $z$, even though the collapsed model accounts for discrete and continuous state dependencies. These results are consistent with those of Zoltowski et al. [2020], where they found Laplace EM compared favorably to both BBVI and particle EM methods. We suspect these results reflect an inherent tradeoff between the fidelity of the posterior approximation and the difficulty of optimization, with simpler approximations (like Laplace EM with the structured mean-field approximation) sometimes leading to improved results [Turner and Sahani, 2011]. We will expand our discussion of these considerations in a camera-ready version.

**Novel modeling contributions (R3).** State space models, such as rSLDS models, form a strong foundation for many types of neural data analysis. However, the ability to easily interpret the interaction between multiple populations within rSLDS models was lacking. We explored three extensions to enhance interpretability, all of which are described within Section 3. Segmenting the continuous latent states for each population (which is equivalent to imposing hard constraints that the $C$ matrix is block diagonal) simply and cleanly allows for per-population states and between-population interactions. On top of that, the "sticky" parameterization of discrete state transitions reveals which neural populations are responsible for staying in, or switching between, discrete states in the model. Finally, we developed further extensions that include more prior information on connectivity, which are discussed in both Section 3 and the Appendix. These contributions also pave the way for further investigation into how structural connectivity could be incorporated into prior distributions on multi-population interactions.

**Findings from the analysis of neural data (R2, R3).** *C. elegans* offers an illustrative demonstration of the mp-srSLDS as there are many possible definitions of 'population'. R3 questioned the value of 1 or 2-neuron populations, but for this organism, this is the level at which neuroscientists frequently study this circuit. The mp-srSLDS naturally handles this limiting case and reveals interactions between neuron classes (Section 5.3; Fig. 4), but it also admits other forms of population structure as well. For example, we explore interactions between ganglia in Appendix C.

Though one might expect strong feedforward influence of PMd on M1 during movement preparation, PMd has been shown to have a weaker influence on M1 during a preparatory phase [Kaufman et al., 2014] compared to during movement, in agreement with our results. Moreover, there are known feedback connections from M1 to PMd to produce the recurrent coupling we see during movement (and which R2 was curious about). More generally, in this dataset, our method allows seeing clear differences between inter-population dynamics during the movement and non-movement states (Section 5.2; Fig. 3), without precisely defining these states *a priori*.

There is a general challenge, shared among all descriptive statistical models, that modeling results do not provide causal insight on brain function. Our goal is that our method can 1) lead to a greater functional understanding, and 2) generate hypotheses that experimental neuroscientists can test with perturbation experiments.

Finally, for a camera-ready version, we will address all the minor concerns, including clarifying figures as suggested and adding missed citations. Thanks again for spending the time to provide valuable feedback on our work.

# References

D. M. Zoltowski, J. W. Pillow, and S. W. Linderman. A general recurrent state space framework for modeling neural dynamics during decision-making. *Proceedings of the 37th International Conference on Machine Learning*, 2020.

R. E. Turner and M. Sahani. Two problems with variational expectation maximisation for time-series models. In *Bayesian Time series models*, chapter 5, pages 109–130. Cambridge University Press, 2011.

M. T. Kaufman, M. M. Churchland, S. I. Ryu, and K. V. Shenoy. Cortical activity in the null space: permitting preparation without movement. *Nature neuroscience*, 17(3):440–448, 2014.


[Meta-Review · NeurIPS 2020]

Four knowledgeable reviewers found that this paper is a solid piece of work. Some concerns were raised about the novelty of the work and the lack of interpretability of the results, however, the reviewers still found that some aspects of the work make it a stepping stone for future research in multi-area interaction. The authors should please try to mitigate these concerns by addressing the ones that can be address in the writing.